# Open data and injuries in urban areas—A spatial analytical framework of Toronto using machine learning and spatial regressions

Eric Vaz[1], Michael D. Cusimano[2], Fernando Bação[3], Bruno Damásio[3]*,
Elissa Penfound[4]

**1** Department of Geography and Environmental Studies, Ryerson University, Toronto, ON, Canada,
**2** Faculty of Medicine, University of Toronto, Toronto, ON, Canada, **3** NOVA IMS Information Management
School, New University of Lisbon, Lisbon, Portugal, **4** Yeates School of Graduate Studies, Ryerson
University, Toronto, ON, Canada

* bdamasio@novaims.unl.pt

## Abstract

Injuries have become devastating and often under-recognized public health concerns. In
Canada, injuries are the leading cause of potential years of life lost before the age of 65. The
geographical patterns of injury, however, are evident both over space and time, suggesting
the possibility of spatial optimization of policies at the neighborhood scale to mitigate injury
risk, foster prevention, and control within metropolitan regions. In this paper, Canada's
National Ambulatory Care Reporting System is used to assess unintentional and intentional
injuries for Toronto between 2004 and 2010, exploring the spatial relations of injury through-
out the city, together with Wellbeing Toronto data. Corroborating with these findings, spatial
autocorrelations at global and local levels are performed for the reported over 1.7 million
injuries. The sub-categorization for Toronto's neighborhood further distills the most vulnera-
ble communities throughout the city, registering a robust spatial profile throughout. Individ-
ual neighborhoods pave the need for distinct policy profiles for injury prevention. This brings
one of the main novelties of this contribution. A comparison of the three regression models
is carried out. The findings suggest that the performance of spatial regression models is sig-
nificantly stronger, showing evidence that spatial regressions should be used for injury
research. Wellbeing Toronto data performs reasonably well in assessing unintentional inju-
ries, morbidity, and falls. Less so to understand the dynamics of intentional injuries. The
results enable a framework to allow tailor-made injury prevention initiatives at the neighbor-
hood level as a vital source for planning and participatory decision making in the medical
field in developed cities such as Toronto.

## 1. Introduction

### 1.1. The injury landscape

Injury is one of the leading causes of death and disability in the United States of America [1].
In Canada alone, an estimated 4.27 million Canadians aged 12 or older, suffered a debilitating

**Data Availability Statement:** Data cannot be
shared as they are owned by a third party and
contain potentially identifiable personal health
information. Data can be requested through the
Canadian Institute for Health Information, under the

Access Data and Reports: https://www.cihi.ca/en/access-data-and-reports.

**Funding:** This research is supported by the Canadian Institutes of Health Research Strategic Team Grant in Applied Injury Research # TIR-103946.

**Competing interests:** The authors have declared that no competing interests exist.

injury between 2009–2010 [2]. The growing number of traumas in urban areas has brought a significant public health concern [3] and fostered a negative perception of health and subjective wellbeing [4]. It is projected that by 2020, injuries will be the third foremost cause of death and disability worldwide [5]. Additionally, the repercussion of injuries from traumatic events has a temporal lag on the psychological and social adjustment of victims, jeopardizing wellbeing in general, and leading to depression [6]. Injuries can be divided into two significant groups generating distinct demographic profiles with leading causes and complex characteristics of epidemiological concern [7]. On one side, unintentional injuries [8] form a leading cause of death in the population between the ages of 1 to 39. Intentional injuries, on the other hand, including assaults and suicides, rank as the second leading cause of death in people ranged 15 to 39. Injuries, therefore, have direct consequences on the active population of Canadians, where three individuals die from injury-related causes every day.

Further to these deaths, fifty Canadians are hospitalized due to injuries [9], which poses a severe strain on the Canadian economy and workplace [10]. Injuries currently represent over seven percent of all hospitalizations [11]. Non-fatal injuries accrue an additional burden to society, as many of these injuries affect the brain or spinal cord, leaving a substantial incidence over permanent disability. Costs on the health-care system in terms of waiting times is evident given the encumbrance over the carrying capacity of hospital systems. Geographical and temporal knowledge of injury events may help in optimizing adequate strategies that convey prevention, control, and efficient monitoring. While until recently, the focus was predominantly on the individual characteristics of the injured person, advances in spatial computation and data science promote new and integrative roles of the spatial aspects of what may lay within the injury landscape at regional level [12–14]. The injury landscape resonates with the concept of regional intelligence [15], where cities may have a proactive role through ubiquitous data integration in mitigating injury risk. By injury landscape, we define the geographical topology of spatially-explicit interactions of injury, where different types of injury occur with particular spatial attributes throughout a given geographic territory.

This paper has the following structure. The next section, Section 1, offers a literature review of the paradigm of injury, and the importance of novel approaches for injury prevention. Section 2 brings the Methodology presenting a systematic framework of the different tools and techniques and exploring the necessary steps of data that allow the statistical and geostatistical analytics. Section 3 discusses the results of the implemented approach for the three regions, and Section 4 offers some concluding remarks and summarizes potential future works.

## 1.2. Literature review

Spatial understanding of the geography of metropolitan areas is of emerging importance in regions that have witnessed rapid urbanization [16], and where the incidence of injuries are positively correlated [17, 18]. Geographic Information Systems (GIS), spatial analysis, and geostatistics allow addressing regional phenomena of health-related concerns in a spatially-explicit context [19]. Several studies analyzed the integration of geographical aspects of public health. For example, Kivell and Mason (1999) used geographic information systems (GIS) to place thirty trauma centers across the United Kingdom [20]. Several authors have also used GIS to predict pedestrian injuries [21–23]. Research on traffic-accident information systems has optimized the capacity to assess the risk of different types of traffic collisions [24–26]. Specifically, in the City of Toronto, researchers have explored the spatial patterns of motor vehicle collisions leading to pedestrian injury based on the pedestrian injury type, age and location within the city [27]. Other studies have examined the relationship between crime and geographic location [28, 29], child maltreatment and geographic location [30], frequency and type

of drug use, which influenced the location of drug and HIV-prevention activities [31], and the likelihood of increased risk of violent injury based on racial segregation [32].

An additional aspect of the spatial patterns of injury that has been explored is the comparison of injury by type in rural versus urban areas. These studies discuss how physical space and subsequent infrastructure (i.e., access and distance to hospitals) links to injury severity and morbidity. Additionally, these studies highlight the importance of understanding the spatial nature of injury by type so that injury prevention strategies may be more accurately targeted [33–35].

Like this study, there are other studies that have explored the spatial nature of injuries with ambulance datasets, including a 2010 study that, with ambulance data from the City of Toronto, explored the spatial and temporal patterns of violent injury [36] and a 2012 study which, with ambulance data, conducted and analysis of outdoor falls based on temporal, spatial and demographic distribution in Laval and Montréal, Canada [37].

The relationship between the spatial distribution of injuries and demographic composition of injured individuals has also been explored. For example, a 2016 study explored the cultural, social and geographic components leading to higher injury risk for Aboriginal peoples in British Columbia, Canada [38]. Another 2016 study explored injury burden caused by accidental venomous bites based on national geography and demographics in Australia [39] and a 2017 study explored the socio- and geo-demographics linked to firearm injuries in Miami-Dade County, Florida [40].

Analysis' of the outcomes of injuries and how they are linked to geographic location and demographics have also been conducted in several studies including a 2019 study which examined the association between injury mortality, geography and sex as it related to youth suicide, senior falls and transport injuries [41]. Furthermore, a study conducted by Keeves and others (2019) used electronic databases of various studies to investigate the outcomes of traumatic injury and their geographic variations, globally. This study found that urban pre-hospital patients have a lower risk of mortality compared to rural patients. This research concludes that there are currently gaps in the literature in regard to determining the link between injury outcome and geography and recommends the use of geographic information systems in future studies related to the spatial distribution of injuries [42].

Despite the many contributions, computational power and data availability have, in recent decades, hindered the opportunity of examining large geographical extents or comparing multiple regions simultaneously. Such studies are particularly important to support regional decision-making for injury prevention proactively and determine key characteristics of injury distributions within urban cores [43–45]. Concise multi-temporal datasets for extensive studies on the injury landscape are rarely available. This study approaches this gap by assessing the complete injury landscape of Toronto. A spatial-analytical framework allows the critical characteristics of different injury types leading to an integrative vision of the consequences and the underlying patterns of injuries in Toronto while benefiting from open data initiatives the city has available. The integration of open data such as Wellbeing Toronto (WT) is addressed at the neighborhood level, offering insights on the potential participatory role of public health initiatives for injury prevention.

## 2. Methodology

### 2.1. Data

**2.1.1. Injury data.** The National Ambulatory Care Reporting System (NACRS) is a comprehensive database that contains demographic, diagnostic, and procedural information on all injury-related occurrences where an ambulance has been dispatched. ICD-10 codes were

**Table 1. Distribution of injury events per main categories.**

| Causes | Total | Percentage |
|---|---|---|
| Injuries from external causes | 1602996 | 93.50% |
| External Morbidity and Mortality | 22888 | 1.33% |
| Intentional Injuries | 1877 | 0.11% |
| Falls | 86751 | 5.06% |

selected for unintentional injuries: (i) resulting from external causes (ICD-10 codes S00 to T14), (ii) external morbidity and mortality (ICD-10 codes V01 to V99), and (iii) fall (ICD-10 codes W00 to W19). For intentional self-harm, the ICD-10 codes from X60 to X84 were used. Data cleaning was carried out further to importing the data from its original format in SAS. The presence of a count with less than five events was discarded and considered as 0. A total of 1714512 injuries (intentional and unintentional) were registered and georeferenced by postal code conversion to latitude and longitude coordinates between 2004 to 2010 (Table 1).

The majority of injuries resulted from external causes of which: (i) injuries to the wrist and hand, (ii) injuries to the head, and (iii) injuries to the knee and lower leg were the most significant cause of ambulance dispatch.

**2.1.2. Socio-economic data.** Wellbeing Toronto (WT) data was used to assess critical variables at the neighborhood level for Toronto (Fig 1). WT corresponds to an integrative and open approach for visualization of Toronto's 140 neighborhoods [46]. As an open data

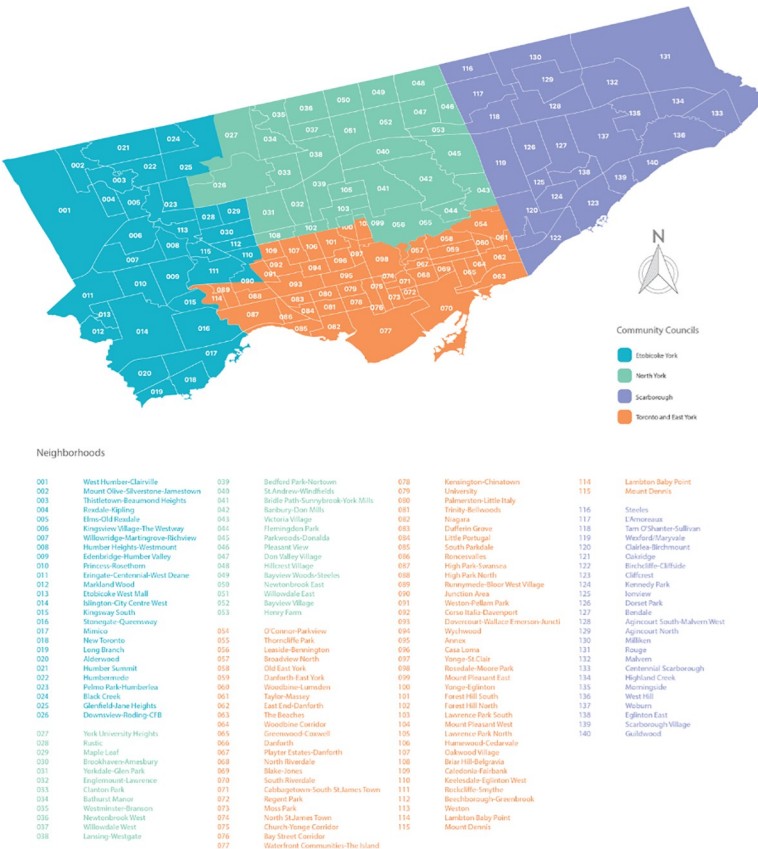

**Fig 1. Toronto neighborhoods** ([1]).

**Table 2. Selected variables from Wellbeing Toronto (N = 140).**

| Variable | Acronym | min | max | mean | sd | Year |
|---|---|---:|---:|---:|---:|---|
| Green Spaces | GreeSp | 0 | 14.271 | 0.58 | 1.29 | 2011 |
| Pollutants Released to Air | PollRel | 0 | 1585690 | 58944.02 | 184007.30 | 2011 |
| Traffic Collisions | TrafCol | 15 | 778 | 173.99 | 123.76 | 2011 |
| Total Population | TotPop | 6577 | 65913 | 19511.22 | 10033.59 | 2014 |
| Low Income Families | LowIncFam | 260 | 10050 | 2184.64 | 1572.53 | 2014 |
| Visible Minority Category | VisMin | 6370 | 65620 | 19226.57 | 9942.22 | 2014 |
| Seniors 65 and over | Sen | 730 | 8990 | 3048.29 | 1579.02 | 2014 |
| Recent Immigrants | RecIm | 95 | 7405 | 1342.75 | 1183.76 | 2014 |
| Low Income Population | LowIncPop | 470 | 15430 | 4164.79 | 3045.62 | 2014 |
| Social Assistance Recipients | SocAssRec | 28 | 5576 | 1385.42 | 1196.94 | 2014 |
| Social Housing Units | SocHous | 0 | 3399 | 641.09 | 653.99 | 2014 |
| Seniors Living Alone | SenLivAl | 40 | 630 | 221.43 | 128.84 | 2014 |
| Rented Dwellings | RentDwell | 200 | 13640 | 3400.68 | 2396.06 | 2014 |
| Drug Arrests | DrugArr | 0 | 174 | 20.76 | 26.47 | 2014 |
| Assaults | Assaults | 9 | 712 | 108.42 | 102.19 | 2014 |
| Robberies | Robberies | 0 | 112 | 20.94 | 20.13 | 2014 |

concept, it hosts a significant amount of data over three reference periods (2008, 2011, and 2014), that include crucial variables encouraging citizen participation, government accountability, and data transparency.

For health analytics, these are vital requisites for successful policy implementation. The Table below shows the variables that were selected from the WT portal (Table 2).

## 2.2. Methods

**2.2.1. Preliminary data organization.** The data was georeferenced utilizing the existing postal code attribute and projected as point features for every single incident onto WGS84. Due to privacy reasons, the data was handled in a secured server and a count selection by location to the nearest census tract performed. This resulted in a generalized geometry dataset. The generalized point count polygons per category of injury were then further simplified onto the neighborhood level and projected into NAD83 17N. The compiled data from WT were added to the data set for further exploration of geostatistical analysis.

**2.2.2. Global spatial autocorrelation.** Global spatial autocorrelation was tested employing a Moran's I index per injury category. This statistic was conducted to test the null hypothesis (Ho) relating to the absence of spatial clustering of injuries in Toronto ($\alpha = 0.05$) (Eq 1):

$$I = \frac{j}{\sum_{i=1}^{i=n}\sum_{i=1}^{i=n} w_{ij}} \cdot \frac{\sum_{i=1}^{i=n}\sum_{j=1}^{j=n} w_{ij}(x_i - x)(x_j - x)}{\sum_{i=1}^{i=n}(x_i - x)^2} \tag{1}$$

Where $w_{ij}$ corresponds to a binary weight matrix defined with the weight of one, given a contiguity of adjacency for any value that holds $w_{ij} = 1$ and any value without adjacency as $w_{ij} = 0$. The product of the distance is defined as $x_i$ for any location $i$ in the distance to relation of its mean. This holds as a statistic for assessing the entire spatial distribution of adjacency formed for the city of Toronto. The null hypothesis was rejected in all categories, suggesting a high spatial autocorrelation for all the injury categories in Toronto.

**2.2.3. Local spatial autocorrelation.** The Local $G_i^*$ statistic was calculated by first determining the injury density. While several approaches allow for spatial density estimation, we

considered that the importance of neighborhood demographics should hold. Thus, the neighborhood injury density results from a ratio where density corresponded to the injuries found in a neighborhood by the population count of the neighborhood. While greater spatial detail could have helped the accuracy of the assessment, one should note that the objective is related to the potential of participatory interaction of injury with available open data. In this sense, neighborhoods are the ideal geographic boundary for governance and city planning.

This approach allowed for the seamless definition of injury density at a spatial level and computation of the statistic, determining the locational aggregation of injury hotspots and cold spots [47]. The calculation of the local $G_i^*$ statistic is as follows (Eq 2):

$$G_i^*(d) = \frac{\sum_{j=1}^{n} w_{i,j} x_{i,j-x} \sum_{j=1}^{n} w_{i,j}}{s\sqrt{\frac{[n\sum_{j=1}^{n} w_{i,j}^2 - (\sum_{j=1}^{n} w_{i,j})^2]}{n-1}}} \tag{2}$$

Where $w_{ij}$ is the spatial weight matrix following a 1 km distance (d), and $w_{ij}(d)$ is assumed as 1. The maps show densities of injury patient residences as hot spots and cold spots, with red representing the highest concentrations of injury and blue the lowest. The selection of regional socio-demographic characteristics for this analysis was guided by previous research and availability of Wellbeing Toronto data.

**2.2.4. Regression framework.** Screening of key demographic variables available at Wellbeing Toronto was carried out by means of a stepwise regression through backward elimination. This allowed for a successful preliminary selection of variables that were applied to three distinct regressions frameworks: (i) spatial lag model, (ii) spatial error model, as well as a non-spatial model to compare performance, and (iii) ordinary least squares model. The spatial lag model (SL) (Eq 3) understands spatial dependency by the addition of a dependent variable that defines the spatial attribute.

$$Y = \rho W y + X\beta + \epsilon, \\ \epsilon \sim N(0, \sigma^2 I) \tag{3}$$

Where I represents an identity matrix, and the $N(0,\sigma^2 I)$ indicates that the errors follow a normal distribution with mean equal to zero and constant variance. When $\rho$ is zero, the lag-dependent term is canceled out, leaving the model under the Ordinary Least Squares (OLS) form. Though when $\rho$ is not zero, it means that spatial dependency exists, and that non-random spatial observable interactions are present [48]. As for the spatial error model (Eq 4), the spatial dependency $\xi$ is accounted within the error term $\epsilon$, assuming the errors of the model as spatially correlated [49].

$$Y = X\beta + \lambda W \xi + \epsilon, \\ \epsilon \sim N(0, \sigma^2 I) \tag{4}$$

## 3. Results

### 3.1. Exploratory data analysis

The Figure below exemplifies the categorization of injuries based on external causes (Fig 2).

Concerning unintentional injuries in Toronto between 2004 and 2010, for the category of external causes, a total of 1602996 were obtained. For external morbidity and mortality, a total number of 22888 were registered, and for falls, a total of 86751 lead to ambulance dispatch. This constituted the larger set of the data used as intentional injuries corresponded only to a fraction of 1877 events, short of 0.12 percent of the total data set.

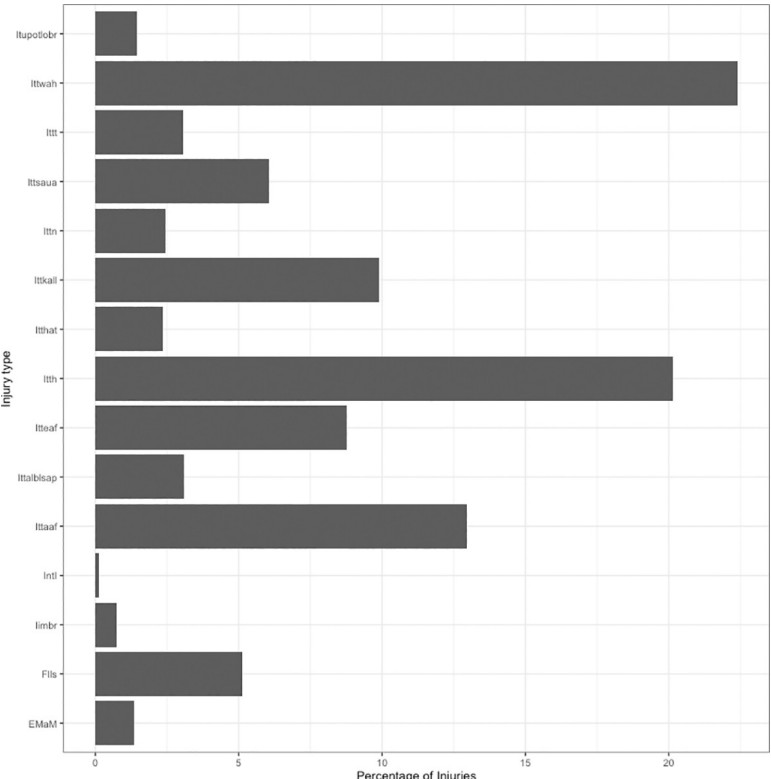

**Fig 2. Percentage of all injury types between 2004 and 2010.** [*] Acronyms: Itth—Injuries to the head; Ittn—Injuries to the neck; Ittt—Injuries to the thorax; Ittalblsap—Injuries to the abdomen, lower back lumbar spine and pelvis; Ittsaua—Injuries to the shoulder and upper arm; Itteaf—Injuries to the elbow and forearm; Ittwah—Injuries to the wrist and hand; Itthat—Injuries to the hip and thigh; Ittkall—Injuries to the knee and lower leg; Ittaaf—Injuries to the ankle and foot; Iimbr—Injuries involving multiple body regions; Itupotlobr—Injuries to unspecified parts of trunk, limb or body region; EMaM—External Morbidity and Mortality; Flls–Falls; IntI—Intentional Injuries.

## 3.2. Spatial autocorrelation

**3.2.1. Global spatial autocorrelation.** Testing for spatial autocorrelation through Moran's I statistic for each event brought evidence that there is significant spatial autocorrelation for all injury categories at the global level (Table 3). Despite regional differences in the rates of unintentional and intentional injuries, the spatial patterns of the residences of those injured by unintentional or intentional mechanisms were found to be highly spatially autocorrelated ($p < 0.01$ for each injury type) indicating that the residence locations of those injured by each of these mechanisms were not randomly distributed in the city of Toronto. This suggests a high spatial clustering that justified further local exploration.

Highest Moran's I values were registered for (a) Injuries to unspecified parts of the trunk, limb or body region, (b) Injuries involving multiple body regions, and (c) Injuries to the neck. While (a) and (b) suggest anatomically more extensive regions, injuries to the neck are quite specific and may become a cause for serious concern given the propensity for physical disability, recovery time, and additional cost to care. The spatial aspects of this injury analysis overall lead to a pressing conclusion that there are clearly geographical determinants that should be assessed to understand the landscape of injury (Table 4).

As expected, all indices remained high, with falls showing very strong spatial-autocorrelation, followed by injuries and injuries leading to mortality. Intentional Injuries had the lowest

**Table 3. Moran's I indices for all categories per neighborhood count.**

| ICD-10 | Injury type | Count | Moran's I | |
|---|---|---|---|---|
| | | | M*i*** | St. Dev |
| S00-S09 | Injuries to the head | 340906 | 0.174 | 9.782 |
| S10-S19 | Injuries to the neck | 41399 | 0.331 | 18.575 |
| S20-S29 | Injuries to the thorax | 52003 | 0.272 | 15.305 |
| S30-S39 | Injuries to the abdomen, lower back, lumbar spine and pelvis | 52273 | 0.292 | 16.414 |
| S40-S49 | Injuries to the shoulder and upper arm | 102450 | 0.275 | 15.420 |
| S50-S59 | Injuries to the elbow and forearm | 148408 | 0.245 | 13.751 |
| S60-S69 | Injuries to the wrist and hand | 378980 | 0.294 | 16.519 |
| S70-S79 | Injuries to the hip and thigh | 39945 | 0.170 | 9.573 |
| S80-S89 | Injuries to the knee and lower leg | 167493 | 0.265 | 14.898 |
| S90-S99 | Injuries to the ankle and foot | 219222 | 0.307 | 17.230 |
| T00-T07 | Injuries involving multiple body regions | 12469 | 0.366 | 20.659 |
| T08-T14 | Injuries to unspecified parts of trunk, limb or body region | 24560 | 0.446 | 25.722 |
| V01-V99 | External Morbidity and Mortality | 22888 | 0.226 | 4.779 |
| W00-W19 | Falls | 86751 | 0.241 | 5.070 |
| X60-X84 | Intentional Injuries | 1877 | 0.158 | 3.465 |

** Significant at the 0.01 confidence level.

Moran's I, however, still corresponding to a very strong Moran's I. Local spatial autocorrelation allows us to assess the neighborhoods at a local scale through the integration of hotspots.

**3.2.2. Local spatial autocorrelation.** The calculation of Local Gi* allowed for the exploration of the spatial distributions of hotspots and their significance levels for the categories of: (i) unintentional injury (external causes), (ii) unintentional injury (resulting in morbidity and mortality), (iii) unintentional injury (due to falls), (iv) intentional injury (self-harm). A weight matrix was generated of queen contiguity type of order 1, for the 140 neighborhoods, as a minimum number of neighbors 3 and a maximum number of neighbors of 11 (Fig 4). The mean and median neighbors corresponded to 5.96 and 6.00, respectively, and a total percentage of non-zero values of 4.26 percent was found.

Fig 3 depicts the Queen contiguity map for neighborhoods in Toronto, nevertheless the most intriguing aspect of these distributions, besides the clear evidence of hotspots and cold spots, was the unique spatial profile of injuries (Fig 4A–4D). Red represents "hotspots", or areas with high injury density, and blue represents cold spots or areas of low or no density of injury. All injury types depict a distinctive pattern.

## 3.3. Regression results

The table below (Table 5) compares the three distinct models. Three of the four injury categories showed moderate performance, suggesting that the data available at Wellbeing Toronto

**Table 4. Moran's I indices for main categories per population distribution.**

| Category | Moran's I |
|---|---|
| Injuries from external causes | 2.918 |
| External Morbidity and Mortality | 2.991 |
| Falls | 5.069 |
| Intentional Injuries | 2.069 |

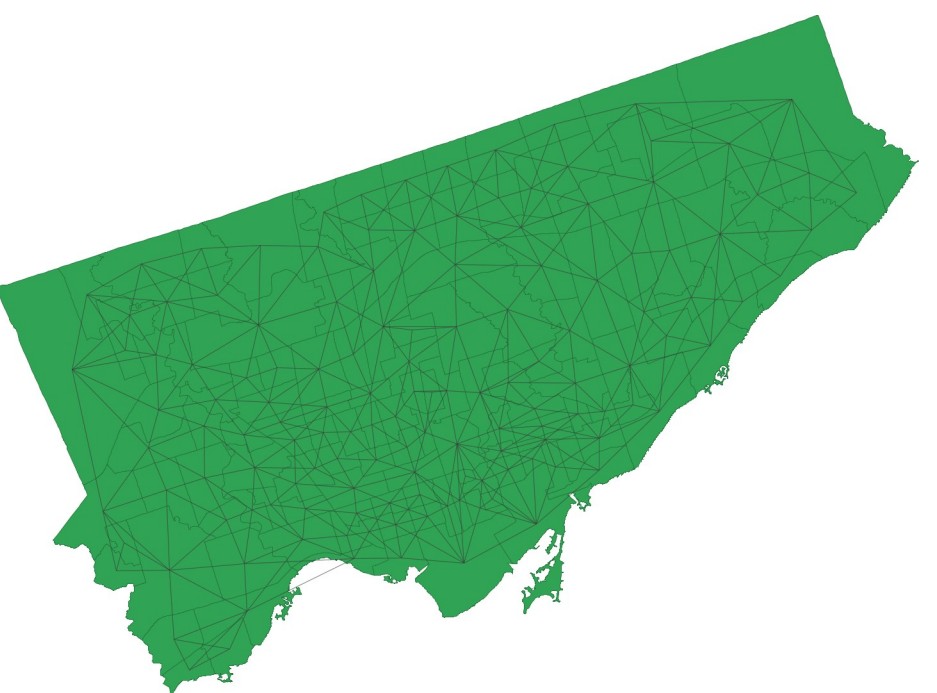

**Fig 3. Queen contiguity map for neighborhoods in Toronto.**

may support well decision making at neighborhood and community participation for injury analysis and integration. In all cases, the spatial regression outperformed the ordinary least squares, with significant improvements in the $r^2$ statistic throughout. The intentional injury model, however, showed low $r^2$, suggesting that demographic data does not explain sufficiently the reasons for self-harm. Finally, it is important to note that injury categories have different variables for each explanatory model, suggesting that there should be different policies and preparedness integration within the city's public health decisions. The following variables were selected through the initial backward elimination as consistent for the models:

i.  Unintentional Injuries: Social Housing, Seniors Living Alone, Total Population, Traffic Collision, Population Density.

ii.  Morbidity: Traffic Collision, Total Population, Visible Minority, Social Housing, Seniors Living Alone, Area (km$^2$).

iii.  Falls: Traffic Collisions, Total Population, Visible Minority, Social Assistant Recipients, Social Housing, Seniors Living Alone, Area (Km$^2$).

iv.  Intentional Injuries: Total Population, Low-Income Families, Low Income Population, Rented Dwellings, Assaults, Robberies, Population Density

## 3.4. SOM cluster results

Analysis of health geography is highly important as it aids in providing evidence of possibly unknown risk factors that may be quantified and better understood only if they are explored spatially [50]. In addition to the regression models (discussed in section 3.3), self-organizing maps (SOM) were built based on the regressors (variables) included in the regression models. In the evaluation of health geography, SOM is a highly useful tool that is used to identify

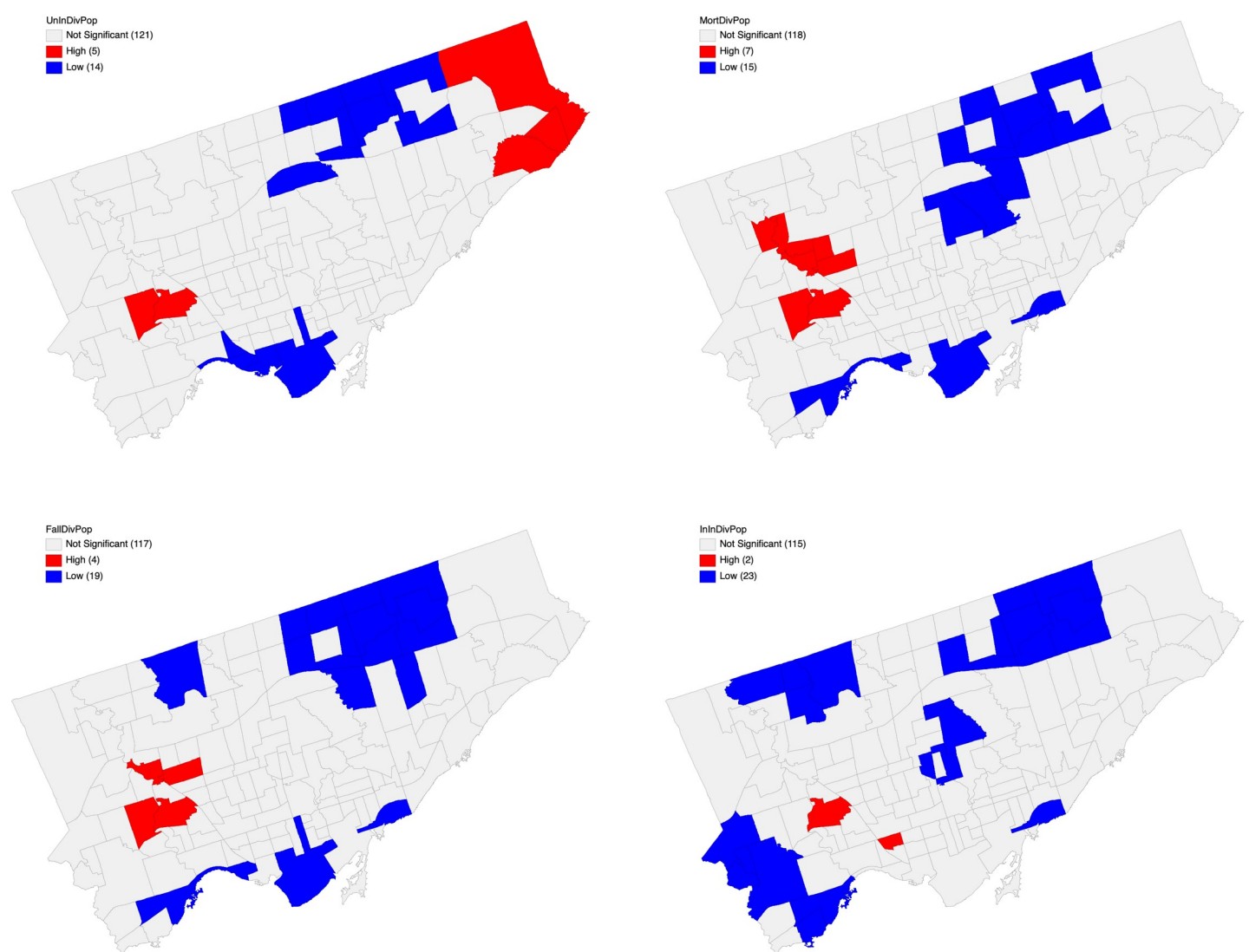

**Fig 4.** a. Unintentional Injury (external causes) Hotspots. b–Unintentional Injury (External Morbidity and Mortality) Hotspots. c–Unintentional Injury (Falls)
Hotspots. d–Intentional Injury Hotspots.

outliers in a dataset [51]. In this analysis, SOM has been used to identify which variables (the attributes or characteristics) are most correlated to injury by type in the City of Toronto neighbourhoods. SOM clusters were generated for each type of injury in the regression model, including unintentional injuries, morbidity, falls and intentional injuries.

The SOM built for unintentional injuries included four clusters. In cluster 1, total population and traffic collisions were the variables that were most strongly correlated with unintentional injuries. This cluster was spatially located in the northeast and northwest peripheries as well as the south-central neighbourhoods in Toronto. In cluster 2, seniors living at home, social housing, and population density were the variables that were most strongly correlated with unintentional injuries. This cluster was spatially distributed throughout Toronto and was less prevalent in south-central Toronto. In cluster 3, traffic collisions, social housing, seniors living at home, total population, population density (ordered from most to least correlated) were the variables that were most strongly correlated with unintentional injuries. This cluster

**Table 5. Comparison of three different regression models (spatial regressions and OLS).**

| Variables | Spatial Lag Model | Spatial Error Model | Ordinary Least Squares |
|---|---|---|---|
| | z-score | z-score | t-statistic |
| **Unintentional Injuries** | | | |
| Social Housing | 1.82714 | 1.49932 | 2.03605 |
| Seniors Living Alone | -2.58747 | -2.23972 | -2.59883 |
| Total Population | -3.04475 | -3.49583 | -2.94316 |
| Traffic Collisions | -1.84367 | -1.7844 | -1.81751 |
| Population Density | -2.87096 | -2.57958 | -3.12535 |
| **R²** | **0.457918** | **0.485750** | **0.421782** |
| **Morbidity** | | | |
| Traffic Collisions | -2.05927 | -2.1162 | -1.92051 |
| Total Population | -2.09607 | -2.52196 | -2.26029 |
| Visible Minority | 1.89049 | 2.30866 | 2.04684 |
| Social Housing | 1.94944 | 1.65653 | 2.29476 |
| Seniors Living Alone | -2.17949 | -1.62644 | -2.29047 |
| Area (Km^2) | 1.90814 | 1.92133 | 1.94006 |
| **R²** | **0.501747** | **0.504612** | **0.480922** |
| **Falls** | | | |
| Traffic Collisions | -2.98132 | -3.01468 | -2.70176 |
| Total Population | -2.19972 | -2.42254 | -2.39975 |
| Visible Minority | 2.03281 | 2.26119 | 2.20993 |
| Social Assistant Recipients | -2.27307 | -2.30639 | -1.96597 |
| Social Housing | 2.43288 | 2.52839 | 2.32207 |
| Seniors Living Alone | -2.32257 | -2.24224 | -1.96219 |
| Area (Km^2) | 2.79554 | 2.6891 | 2.70456 |
| **R²** | **0.528434** | **0.527985** | **0.507562** |
| **Intentional Injuries** | | | |
| Total Population | -2.87985 | -2.64276 | -2.91692 |
| Low-Income Families | -1.33643 | -1.27004 | -1.46417 |
| Low Income Population | 1.68586 | 1.53451 | 1.72947 |
| Rented Dwellings | -2.36279 | -2.2732 | -2.13923 |
| Assaults | 3.11778 | 2.90505 | 3.06752 |
| Robberies | -2.90614 | -2.90247 | -2.69822 |
| Population Density | 3.50405 | 3.43355 | 3.3855 |
| **R2** | **0.296519** | **0.290263** | **0.269535** |

was also spatially distributed throughout Toronto but was more prevalent in south-central Toronto. In cluster 4, population density was most strongly correlated with unintentional injuries. This cluster was represented in a single neighbourhood, located in Toronto's city center. The results of the heatmaps for the regressors (for unintentional injuries) have been summarized in Table 6, which shows a breakdown of the cluster that each variable is highly correlated with. These results show that clusters 1 and 2 have the highest number of variables correlated with unintentional injuries, whereas cluster 4 has fewer variables correlated with unintentional injuries and cluster 3 has variables that are only moderately correlated with unintentional injuries.

The SOM built for morbidity also included four clusters. In cluster 1, seniors living at home and social housing were the variables that were most strongly correlated with morbidity. This cluster was spatially distributed throughout Toronto but was less prevalent in the northeast

**Table 6. Unintentional injuries heatmaps for the regressors summary.**

| Unintentional Injuries | |
|---|---|
| **Variable** | **Cluster** |
| Social Housing | 2 |
| Seniors Living Alone | 2 |
| Total Population | 1 |
| Traffic Collisions | 1 |
| Population Density | 4 |

part of the city. In cluster 2, seniors living at home, traffic collisions total population and visible minorities (ordered from most to least correlated) were the variables that were most strongly correlated with morbidity. This cluster was spatially distributed throughout Toronto but was more prevalent in the north, south, central and southwest. In cluster 3, total population and visible minorities were the variables most strongly correlated with morbidity, however, seniors living at home was also strongly correlated with morbidity. This cluster was represented in only two neighbourhoods, one is south-central and the other in east Toronto. In cluster 4, traffic collisions and area were most strongly correlated with morbidity. This cluster was also only represented in two neighbourhoods, one in northeast and the other in northwest Toronto. The results of the heatmaps for the regressors (for morbidity) have been summarized in Table 6, which shows a breakdown of the cluster that each variable is highly correlated with. These results show that clusters 3 and 4 have the highest number of variables correlated with morbidity, whereas clusters 1 and 2 have fewer variables correlated with morbidity.

The SOM built for falls, again, included 4 clusters. In cluster 1 seniors living alone, social assistance recipients, social housing, traffic collisions, total population, and visible minorities (ordered from most to least correlated) were the variables most strongly correlated with falls. This cluster was distributed throughout Toronto and was the dominating cluster, representing the majority of the city. In cluster 2, social housing was most strongly correlated with falls. This cluster was only represented in two neighbourhoods, both located in south-central Toronto. In cluster 3, total population and visible minorities were most strongly correlated with falls. This cluster was also only represented in two neighbourhoods, one located in south-central and the other located in east Toronto. In cluster 4, traffic collisions and area were the variables most strongly correlated with falls. Like clusters 2 and 3, this cluster was also only represented in two neighbourhoods, one in northeast and the other in northwest Toronto. The results of the heatmaps for the regressors (for falls) have been summarized in Table 7, which shows a breakdown of the cluster that each variable is highly correlated with. These results

**Table 7. Falls heatmaps for the regressors summary.**

| Falls | |
|---|---|
| **Variable** | **Cluster** |
| Traffic Collisions | 4 |
| Total Population | 3 |
| Visible Minority | 3 |
| Social Assistance Programs | 1 |
| Social Housing | 2 |
| Seniors Living Alone | 1 |
| Area | 4 |

**Table 8. Intentional injuries heatmaps for the regressors summary.**

| Intentional Injuries | |
|---|---|
| Variable | Cluster |
| Total Population | 1 and 2 |
| Low Income Families | 1 and 2 |
| Low Income Population | 1 |
| Rented Dwellings | 2 and 3 |
| Assaults | 2 |
| Robberies | 2 |
| Population Density | 3 |

show that clusters 1, 3 and 4 have the highest number of variables correlated with falls, whereas cluster 2 has fewer variables correlated with falls.

The SOM built for intentional injuries only included 3 clusters (Table 8). In cluster 1, low-income population, low-income family, total population, rented dwelling, and population density (ordered from most to least correlated) were strongly correlated with intentional injuries. This cluster was distributed throughout Toronto and was the dominating cluster, representing most neighbourhoods in the city. In cluster 2, rented dwelling, robberies, assaults, low-income population, low-income family, total population (ordered from most to least correlated) were the variables strongly correlated with intentional injuries. This cluster was represented in several neighbourhoods, all spatially located in south central Toronto. In cluster 3, rented dwelling and population density were strongly correlated with intentional injuries. This cluster was only represented in two neighbourhoods, both located in central Toronto. The results of the heatmaps for the regressors (for intentional injuries) have been summarized in Table 9, which shows a breakdown of the cluster that each variable is highly correlated with. These results show that cluster 2 has the highest number of variables correlated with unintentional injuries, whereas clusters 1 and 3 have fewer variables correlated with intentional injuries.

Seniors living alone and traffic collisions were strongly correlated with the majority of clusters for unintentional injuries, morbidity, and falls. Indicating that these variables may be more likely to contribute to these types of injuries compared to the other variables included in this analysis. Rented dwelling and low-income population were strongly correlated with the clusters for intentional injuries, indicating that intentional injuries are more likely to occur in poorer (or low income) Toronto neighbourhoods. Overall, clusters that were represented by a larger number of neighbourhoods tended to have a higher number of variables correlated with each injury type, while smaller clusters tended to have fewer numbers of (or more specific) variables associated with injury type. Population density and rented dwellings were variables that

**Table 9. Morbidity heatmaps for the regressors summary.**

| Morbidity | |
|---|---|
| Variable | Cluster |
| Traffic Collisions | 4 |
| Total Population | 3 |
| Visible Minority | 3 |
| Social Housing | 1 |
| Seniors Living Alone | 2 |
| Area | 4 |

tended to be associated with locations in central Toronto (i.e., the neighbourhoods that have higher population density compared to the city's peripheries).

## 4. Conclusions

Recent advances in geocomputational methods, as well as spatial analysis, have brought new techniques that better enable the understanding of spatial characteristics of cities and regions [52]. It is of utmost importance to understand regional patterns of epidemiologic concern, to better optimize public health efficiency in rapidly changing regions [53]. In this sense, geocomputational methods, when combined with large spatially-explicit data, allow for significant contributions of regional understanding of injury dynamics. Supported by data availability, open data at the city level may have a profound impact on the assessment and resulting community and policy intervention strategies for neighborhoods. The application of geocomputational techniques to the National Ambulatory Care Reporting System has allowed perceiving that the pattern of the residence locations of injured persons is not spatially random, but clearly very spatially dependent.

There is some disagreement in the literature regarding the effects of immigration status on health and violence. A number of authors have shown that population health determinants such as income and social status, education, employment or working conditions, social and physical environments, personal health practices, healthy child development, biologic and genetic endowment, health services, sex, and culture have a relationship with injury patterns [54–56]. Others have argued that the distinction between intentional and unintentional injury is arbitrary [57, 58] and that the risk factors associated with intentional and unintentional injury overlap [59–61]. Based on these lines of previous work, we would have expected that the spatial distributions of the residences of those injured by these disparate mechanisms may have overlapped. However, ours is the first study to demonstrate that the spatial distributions of residence locations were similar regardless of whether the mechanism of injury was intentional or unintentional. This finding was consistently seen in the choice of selected variables, despite marked differences in size, economy, and cultural composition. The slightly larger areas of hotspots of home locations of those injured unintentionally may either reflect a difference of the aforementioned factors or simply be related to the larger number of persons injured unintentionally. Finally, the most resounding conclusion is that injury can greatly benefit from tailor-made injury prevention initiatives that address the specificities of neighborhoods and types of injury to guarantee a successful mechanism of injury prevention at the local level.

## Author Contributions

**Conceptualization:** Eric Vaz, Michael D. Cusimano.

**Formal analysis:** Eric Vaz.

**Investigation:** Michael D. Cusimano.

**Methodology:** Eric Vaz, Michael D. Cusimano, Fernando Bação, Bruno Damásio.

**Software:** Bruno Damásio.

**Validation:** Eric Vaz.

**Visualization:** Eric Vaz, Bruno Damásio, Elissa Penfound.

**Writing – original draft:** Eric Vaz, Michael D. Cusimano, Fernando Bação, Bruno Damásio, Elissa Penfound.

**Writing – review & editing:** Eric Vaz, Michael D. Cusimano, Fernando Bação, Bruno Damásio, Elissa Penfound.

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
