## [Decision Letter · Decision Letter 0]

4 Jan 2021

PONE-D-20-32866

Open Data and Injuries in the urban areas – A spatial analytical framework of Toronto using machine learning and spatial regressions

PLOS ONE

Dear Dr. Damásio,

Thank you for submitting your manuscript to PLOS ONE. After careful consideration, we feel that it has merit but does not fully meet PLOS ONE’s publication criteria as it currently stands. Therefore, we invite you to submit a revised version of the manuscript that addresses the points raised during the review process.

We look forward to receiving your revised manuscript.

Kind regards,

Yanyong Guo, Ph.D

Academic Editor

PLOS ONE

Journal Requirements:

2.We suggest you thoroughly copyedit your manuscript for language usage, spelling, and grammar. If you do not know anyone who can help you do this, you may wish to consider employing a professional scientific editing service.  

3.In your Data Availability statement, you have not specified where the minimal data set underlying the results described in your manuscript can be found. PLOS defines a study's minimal data set as the underlying data used to reach the conclusions drawn in the manuscript and any additional data required to replicate the reported study findings in their entirety. All PLOS journals require that the minimal data set be made fully available. For more information about our data policy, please see http://journals.plos.org/plosone/s/data-availability.

4.Thank you for stating the following in the Acknowledgments Section of your manuscript:

"This research is supported by the Canadian Institutes of Health Research Strategic Team Grant in

Applied Injury Research # TIR-103946."

 "NO - The funders had no role in study design, data collection and analysis, decision to publish, or preparation of the manuscript."

5. Please ensure that you refer to Figure 3 in your text as, if accepted, production will need this reference to link the reader to the figure.

6.We note that the figures in your submission contain map images which may be copyrighted. All PLOS content is published under the Creative Commons Attribution License (CC BY 4.0), which means that the manuscript, images, and Supporting Information files will be freely available online, and any third party is permitted to access, download, copy, distribute, and use these materials in any way, even commercially, with proper attribution. For these reasons, we cannot publish previously copyrighted maps or satellite images created using proprietary data, such as Google software (Google Maps, Street View, and Earth). For more information, see our copyright guidelines: http://journals.plos.org/plosone/s/licenses-and-copyright.

1.    You may seek permission from the original copyright holder of the figures to publish the content specifically under the CC BY 4.0 license. 

7. We note that you currently have two different versions of tables 1 to 4 included in your manuscript, so that the tables can be differentiated can you please update the table titles (numbering) and in-text citations so that the second set of tables are numbered individually (and not the same as the previous numbers for Tables 1 to 4 already used).

Reviewers' comments:

Reviewer's Responses to Questions

**Comments to the Author**

1. Is the manuscript technically sound, and do the data support the conclusions?

Reviewer #1: Yes

Reviewer #2: Yes

Reviewer #3: Partly

2. Has the statistical analysis been performed appropriately and rigorously? 

Reviewer #1: Yes

Reviewer #2: Yes

Reviewer #3: Yes

3. Have the authors made all data underlying the findings in their manuscript fully available?

Reviewer #1: Yes

Reviewer #2: Yes

Reviewer #3: Yes

4. Is the manuscript presented in an intelligible fashion and written in standard English?

Reviewer #1: Yes

Reviewer #2: Yes

Reviewer #3: Yes

5. Review Comments to the Author

Reviewer #1: The topic of this paper is interesting and important. The methods sound. The results are meaningful and useful. There is one suggestion to improve this paper.

1. The quality of the figures are not high. And the figure name and figure number need to be with the figures.

Reviewer #2: It is not surpervise that the ML approach could achive a better results. The motivation should be strengthened. The contributions of the study should be clear. Some related references should be discussed. in addition, there are some typos.

Reviewer #3: The manuscript aims at exploring the spatial relations of injury throughout the city, together with Wellbeing Toronto data. Given the high attention of urban injuries, this work is of great significance. However, there are still some major problems in the current manuscript, and should be addressed.

(1) why use the 2004-2010 injuries data, why not use the recent, do you make sure that results of this study can be applied for the nowadays?

(2) Covariates used in this study are not consistent in the date, which may affect the reliability of findings.

(3) literature reviews are too simple, suggest to added more relevant researches.

(4) figures in this current manuscript are unclear.

(5) table 2 are too simple, some statistics should be done, such as mean, s.d. max and min.

(6) The label of the table is too messy, please check.

(7) please add the line number, it is very inconvenient for reviewers.

(8) Grammatical errors can be seen in several sentences. The whole manuscript needs to be thoroughly proofread.

6. PLOS authors have the option to publish the peer review history of their article (what does this mean?). If published, this will include your full peer review and any attached files.

Reviewer #1: No

Reviewer #2: No

Reviewer #3: No

---

## [Author Response · Author response to Decision Letter 0]

11 Feb 2021

Reviewer 1

Reviewer #1: The topic of this paper is interesting and important. The methods sound. The results are meaningful and useful. There is one suggestion to improve this paper.

1. The quality of the figures are not high. And the figure name and figure number need to be with the figures.

We’ve changed Figure 1, and certified that all Figures are at 300 dpi for easy visualization upon the final version. Finally, we’ve added an acknowledgment section thanking the reviewer for the inspiring comments concerning our manuscript. Thank you very much for your review!

Reviewer 2

Reviewer #2: It is not surpervise that the ML approach could achive a better results. The motivation should be strengthened. The contributions of the study should be clear. Some related references should be discussed. in addition, there are some typos.

Response: Thank you. We’ve now strengthened the motivation, and proofread the manuscript. We would like to thank the reviewer for the inspiring comment, an acknowledgment section has been added accordingly. 

Reviewer 3

Reviewer #3: The manuscript aims at exploring the spatial relations of injury throughout the city, together with Wellbeing Toronto data. Given the high attention of urban injuries, this work is of great significance. However, there are still some major problems in the current manuscript, and should be addressed.

Query 1: why use the 2004-2010 injuries data, why not use the recent, do you make sure that results of this study can be applied for the nowadays.

Response 1: NACRS data took circa ten years to get governmental and institutional approval. This is a long process itself. We are convinced that the fact that we have assessed a total of six years timeframe, renders this model current, given that no significant changes have occurred within the dynamics of the city that would radically change the predictive capacity of the response variables. 

Query 2: Covariates used in this study are not consistent in the date, which may affect the reliability of findings.

Response 2: While we do understand the reviewers concern, we would like to remind the reviewer that we are using cumulative data between 2004 and 2010, approximate values within the same timeline for covariates are common practice in social sciences as both census, and demographic data are not necessarily yearly derived. Furthermore, they are used for an explanatory model of the phenomena, and not for a direct comparison of an output. 

Query 3: literature reviews are too simple, suggest to added more relevant researches.

Response 3: Thanks. We’ve added additional relevant research. Please refer to discussion from line numbers 145 to 175. Thanks. 

Query 4: figures in this current manuscript are unclear.

Response 4: We’ve changed Figure 1. Additionally, we’ve reformatted Figure 2 to show the relative percentage of injuries, reflecting better the distribution of injury. The other figures are now at 300 dpi as to guarantee higher resolution both for online and print visualization. 

Query 5: table 2 are too simple, some statistics should be done, such as mean, s.d. max and min. 

Response 5: Thanks. We’ve now added these. Please refer to the new Table 2.

Query 6: The label of the table is too messy, please check.

Response 6: We’ve now revised the labels of Table 2. 

Query 7: please add the line number, it is very inconvenient for reviewers.

Response 7: Line numbers have been added. 

Query 8: Grammatical errors can be seen in several sentences. The whole manuscript needs to be thoroughly proofread.

Response 8: Thanks. A thorough proofreading of the manuscript was now conducted. 

Acknowledgment: We would like to thank the reviewer for the useful comments and critique that greatly improved the manuscript. We’ve added an acknowledgment section accordingly. Thank you very much!

---

## [Decision Letter · Decision Letter 1]

24 Feb 2021

Open Data and Injuries in the urban areas – A spatial analytical framework of Toronto using machine learning and spatial regressions

PONE-D-20-32866R1

Dear Dr. Damásio,

We’re pleased to inform you that your manuscript has been judged scientifically suitable for publication and will be formally accepted for publication once it meets all outstanding technical requirements.

Kind regards,

Yanyong Guo, Ph.D

Academic Editor

PLOS ONE

Additional Editor Comments (optional):

Reviewers' comments:

Reviewer's Responses to Questions

**Comments to the Author**

1. If the authors have adequately addressed your comments raised in a previous round of review and you feel that this manuscript is now acceptable for publication, you may indicate that here to bypass the “Comments to the Author” section, enter your conflict of interest statement in the “Confidential to Editor” section, and submit your "Accept" recommendation.

Reviewer #1: (No Response)

Reviewer #3: All comments have been addressed

2. Is the manuscript technically sound, and do the data support the conclusions?

Reviewer #1: (No Response)

Reviewer #3: Yes

3. Has the statistical analysis been performed appropriately and rigorously? 

Reviewer #1: (No Response)

Reviewer #3: Yes

4. Have the authors made all data underlying the findings in their manuscript fully available?

Reviewer #1: (No Response)

Reviewer #3: Yes

5. Is the manuscript presented in an intelligible fashion and written in standard English?

Reviewer #1: (No Response)

Reviewer #3: Yes

6. Review Comments to the Author

Reviewer #1: (No Response)

Reviewer #3: (No Response)

7. PLOS authors have the option to publish the peer review history of their article (what does this mean?). If published, this will include your full peer review and any attached files.

Reviewer #1: No

Reviewer #3: No

---

## [Editor Report · Acceptance letter]

26 Feb 2021

PONE-D-20-32866R1 

Open Data and Injuries in urban areas – A spatial analytical framework of Toronto using machine learning and spatial regressions 

Dear Dr. Damásio:

I'm pleased to inform you that your manuscript has been deemed suitable for publication in PLOS ONE. Congratulations! Your manuscript is now with our production department. 

Kind regards, 

on behalf of

Dr. Yanyong Guo 

Academic Editor

PLOS ONE